# We're Not Using Videos Effectively:
# An Updated Domain Adaptive Video Segmentation Baseline

**Simar Kareer, Vivek Vijaykumar, Harsh Maheshwari, Prithvijit Chattopadhyay,**
**Judy Hoffman, Viraj Prabhu**
*Georgia Institute of Technology*
*Correspondence: skareer@gatech.edu*

**Reviewed on OpenReview:** *https://openreview.net/forum?id=1OR6iX6JHm*

## Abstract

There has been abundant work in unsupervised domain adaptation for semantic segmentation (DAS) seeking to adapt a model trained on images from a labeled source domain to an unlabeled target domain. While the vast majority of prior work has studied this as a frame-level Image-DAS problem, a few *Video-DAS* works have sought to additionally leverage the temporal signal present in adjacent frames. However, Video-DAS works have historically studied a distinct set of benchmarks from Image-DAS, with minimal cross-benchmarking. In this work, we address this gap. Surprisingly, we find that (1) even after carefully controlling for data and model architecture, state-of-the-art Image-DAS methods (HRDA and HRDA+MIC) outperform Video-DAS methods on established Video-DAS benchmarks (+14.5 mIoU on Viper→Cityscapes-Seq, +19.0 mIoU on Synthia-Seq→Cityscapes-Seq), and (2) naive combinations of Image-DAS and Video-DAS techniques only lead to marginal improvements across datasets. To avoid siloed progress between Image-DAS and Video-DAS, we open-source our codebase with support for a comprehensive set of Video-DAS and Image-DAS methods on a common benchmark. Code available at this link.

## 1 Introduction

Deep learning systems perform remarkably well when training and test data are drawn from the same distribution. However, this assumption is often violated in the real world *e.g.* under changing weather (Hoffman et al., 2018), geographies (Prabhu et al., 2022; Kalluri et al., 2023), or curation strategies (Saenko et al., 2010), wherein annotating additional data may be prohibitively expensive. To address this, unsupervised domain adaptation (UDA) (Saenko et al., 2010; Ganin & Lempitsky, 2015; Tzeng et al., 2017) attempts to train a model which performs well on the target domain given access to labeled source data and unlabeled target data. UDA has seen successful application to computer vision tasks such as semantic segmentation (Hoffman et al., 2018; Hoyer et al., 2022b;c), object detection (Chen et al., 2018; Li et al., 2022; Vibashan et al., 2023) and person re-identification (Mekhazni et al., 2020; Zhao et al., 2021). In this work, we focus on the complex task of semantic segmentation adaptation due to its high labeling cost and ubiquity in practical applications such as self-driving (Wu et al., 2021).

Since data from simulation can be cheaply annotated (Richter et al., 2016), several works for se-

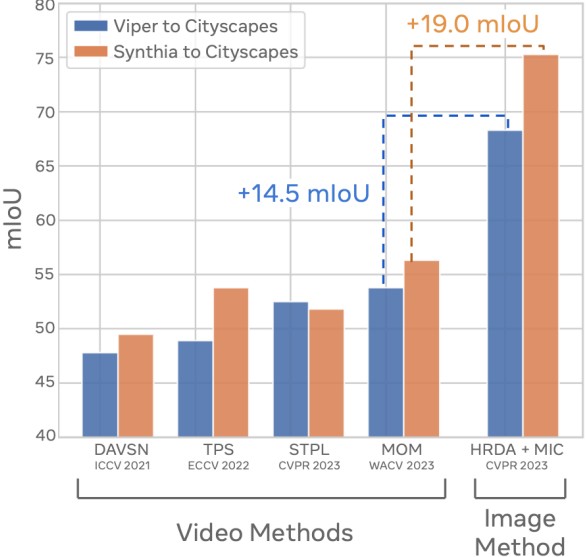

Figure 1: **Overview.** Recent domain adaptive video segmentation methods do not compare against state-of-the-art baselines for Image-DAS. We perform the first such cross-benchmarking, and find that even after controlling for data and model architecture, Image-DAS methods *strongly* outperform Video-DAS methods on the two key Video-DAS benchmarks.

mantic segmentation adaptation have attempted to leverage *labeled simulated* data (Hoyer et al., 2022b) to adapt to unlabeled real data. Benchmarks for this task are commonly in the context of autonomous driving, where the source data are *videos* from a driving simulator, and the target data are *videos* from a real dashcam. This problem has seen considerable progress: for instance, on the popular GTAV (Richter et al., 2016)→Cityscapes (Cordts et al., 2016) benchmark, absolute performance has jumped by >45 points since its inception (Hoffman et al., 2016). Surprisingly though, the works driving this progress have typically ignored the sequential nature of the data and studied this as an Image-DAS problem.

A few isolated works have pioneered the problem of Video-DAS and attempt to leverage temporal dynamics in domain alignment for semantic segmentation (Guan et al., 2021; Shin et al., 2021; Xing et al., 2022; Wu et al., 2022; Cho et al., 2023; Lo et al., 2023). However, these works employ a different set of benchmarks and methods than the Image-DAS methods. As a result, progress on these two related problems has been siloed, with no recent cross-benchmarking. We bridge this gap and present updated baselines for Video-DAS.

Concretely, we benchmark the performance of HRDA (Hoyer et al., 2022b) and HRDA+MIC (Hoyer et al., 2022c), state-of-the-art methods for Image-DAS, on Video-DAS benchmarks. We find that even after carefully controlling for model architecture and training data, HRDA+MIC outperforms state-of-the-art Video-DAS methods, *e.g.* by 14.5 mIoU on Viper→Cityscapes-Seq, and 19.0 mIoU on Synthia-Seq→Cityscapes-Seq (Figure 1), the two established benchmarks for this task. We perform an ablation study to identify the source of this improvement and find multi-resolution fusion (Hoyer et al., 2022b) to be the most significant factor. In essence, multi-resolution fusion enables segmentation models to utilize full resolution images rather than down-sampling, while maintaining a manageable memory footprint.

The strong performance of recent Image-DAS methods (e.g. HRDA+MIC) on Video-DAS benchmarks raises the question of whether current Video-DAS methods are still useful given this updated baseline. To study this, we benchmark a comprehensive set of Video-DAS techniques from the literature, including a video-level domain discriminator, the ACCEL video segmentation architecture (Jain et al., 2019), temporally consistent mix-up (Cho et al., 2023), and a suite of pseudo-label refinement strategies. We exhaustively test combinations of these methods with HRDA and do not observe significant gains with any single combination. In contrast to prior work which only reports on Viper→Cityscapes-Seq and Synthia-Seq→Cityscapes-Seq, we propose and corroborate these findings on two additional Video-DAS shifts in order to draw stronger conclusions. Specifically, we propose Viper→BDDVid and Synthia-Seq→BDDVid, where BDDVid is constructed from BDD10k (Yu et al., 2020).

In the process, we develop and open-source `UnifiedVideoDA`, the first modern codebase built on top of a popular semantic segmentation library, MMSegmentation (Contributors, 2020), which supports a range of Image-DAS and Video-DAS techniques out-of-the-box. We hope that this codebase will bridge the two communities and bolster an increased exchange of ideas.

We make the following contributions:

- We discover that even after controlling for model architecture, HRDA+MIC (a recently proposed Image-DAS method) sets the state-of-the-art on existing Video-DAS benchmarks, outperforming specialized Video-DAS methods, by as much as +14.5 mIoU on Viper→Cityscapes-Seq and +19.0 mIoU on Synthia-Seq→Cityscapes-Seq.

- We perform an ablation study to explain this performance gap, and discover multi-resolution fusion (Hoyer et al., 2022b), which enables utilization of full-resolution images with a manageable memory footprint, to be the largest contributing factor.

- We explore hybrid UDA strategies which add popular Video-DAS techniques to our updated baseline. While a few selectively improve performance, *e.g.* pseudo-label refinement (+4.6 mIoU, Table 6), and targeted architectural modification (+2.2 mIoU, Table 4), we do not find any strategy with significant improvements across existing benchmarks and two additional shifts we propose.

- We open-source our codebase `UnifiedVideoDA`, built on top of MMSegmentation, a commonly used library for Image-DAS. We enable Video-DAS research in this codebase by enabling it to load consecutive frames, optical flow, and implementing baselines for key video techniques. We hope this codebase can help the community push the frontier of Video-DAS and Image-DAS simultaneously.

## 2 Related work

### 2.1 Image-level UDA for Semantic Segmentation (Image-DAS)

Approaches for unsupervised domain adaptive semantic segmentation typically leverage one (or both) of two techniques: distribution matching and self-training. Distribution matching-based methods attempt to align source and target domains in a shared representation space via techniques such as domain adversarial learning (Hoffman et al., 2016; Tzeng et al., 2017; Tsai et al., 2018; Vu et al., 2019). Some works additionally perform style transfer from source to target and ensure predictive consistency (Murez et al., 2017; Hoffman et al., 2018). In contrast, self-training approaches iteratively generate pseudo-labels for unlabeled target examples from a source model and retrain on these. A number of techniques are used to improve the quality of these pseudo-labels, such as using prediction confidence (Prabhu et al., 2021b; Tan et al., 2020; Mei et al., 2020), augmentation consistency (Prabhu et al., 2021a; Melas-Kyriazi & Manrai, 2021), student-teacher learning (Tarvainen & Valpola, 2018) and cross-domain mixing (Tranheden et al., 2020). The state-of-the-art method for this problem (Hoyer et al., 2022c;b;a) combines such pseudo-label-based self-training, with a SegFormer (Xie et al., 2021) backbone, in addition to auxiliary objectives for feature distance regularization and rare class sampling (Hoyer et al., 2022a), multi-resolution fusion (Hoyer et al., 2022b) and masked prediction (Hoyer et al., 2022c). In this work, we cross-benchmark these methods on Video-DAS datasets.

### 2.2 Video-level Unsupervised Domain Adaptation

Most prior works for video UDA are designed for the tasks of semantic segmentation and action recognition, and we now survey both these lines of work.

**Semantic Segmentation.** Given the strong progress in UDA for semantic segmentation, a few works have explored how to integrate video information to improve performance, which we refer to as Video-DAS. DAVSN (Guan et al., 2021) was the first work to tackle this problem, and established two benchmarks: Viper to Cityscapes-Seq and Synthia-Seq to Cityscapes-Seq (Richter et al., 2017; Ros et al., 2016; Cordts et al., 2016), along with a method based on temporal consistency regularization realized by combining adversarial learning and self-training. A few works have since built upon this. Xing *et al.* (Xing et al., 2022) encourages spatio-temporal predictive consistency via a cross-frame pseudo-labeling strategy. Wu *et al.* (Wu et al., 2022) also encourages temporal consistency, and includes an image level domain discriminator. Shin *et al.* (Shin et al., 2021) uses a two-stage objective of video adversarial learning followed by video self-training, whereas Cho *et al.* (Cho et al., 2023) optimizes a temporally consistency cross-domain mixing objective, which aligns representations between the mixed and source domains. Finally, Lo *et al.* (Lo et al., 2023) uses spatio-temporal pixel-level contrastive learning for domain alignment. We summarize the differences of key methods in Table 1, and benchmark against all methods in Table 2.

**Action Recognition.** Video UDA for action recognition has been studied extensively, and broadly falls under four categories (Xu et al., 2022): Adversarial, discrepancy-based, semantic-based, and reconstruction-based. Adversarial methods such as (Chen et al., 2019; 2022b; Pan et al., 2019) use video-level domain discriminators to align source and target domains, and aligns features at different levels by focusing on key parts of the videos, for instance by introducing temporal co-attention to attend over key video frames (Pan et al., 2019). Other works explore alignment across additional modalities, such as optical flow (Munro & Damen, 2020) and audio (Yang et al., 2022). In contrast, discrepancy-based methods (Jamal et al., 2018; Gao et al., 2020) explicitly minimize feature distances between source and target video clips, while semantic-based methods employ contrastive objectives by creating positive pairs from fast and slow versions of the same video, cutmix augmentations of a given clip (Sahoo et al., 2021), or by leveraging auxiliary modalities (Song et al., 2021) (negatives are randomly sampled from other other videos or by shuffling frames). Finally, reconstruction-based approaches (Wei et al., 2023) learn to reconstruct video clips and separate task-specific features from domain-specific ones in the learned latent space.

Unfortunately, progress in Video-DAS and Image-DAS has been siloed, with each using a different and incomparable set of benchmarks, backbones, and algorithms. We address this gap by cross-benchmarking state-of-the-art Image-DAS techniques on Video-DAS benchmarks, and additionally investigate the effectiveness of algorithms that combine techniques from both lines of work.

Table 1: Characteristics of prior works in Video-DAS. While these works have additional components, we highlight the components which utilize sequential video frames.

| Approach | Video Discrim. | ACCEL | Consis. Mixup | PL. Refine |
|---|:---:|:---:|:---:|:---:|
| DAVSN (Guan et al., 2021) | ✓ | ✓ | | MaxConf |
| UDA-VSS (Shin et al., 2021) | ✓ | | | Consis |
| TPS (Xing et al., 2022) | | ✓ | | Warp Frame |
| I2VDA (Wu et al., 2022) | | | | Custom |
| Moving Object Mixup (Cho et al., 2023) | | ✓ | ✓ | Consis |

## 3 Preliminaries

### 3.1 Video UDA for Semantic Segmentation (Video-DAS)

In unsupervised domain adaptation (UDA), the goal is to adapt a model trained on a labeled source domain to an unlabeled target domain. Formally, given labeled source examples $(x_\mathcal{S}, y_\mathcal{S}) \in D_\mathcal{S}$ and unlabeled target examples $x_\mathcal{T} \in D_\mathcal{T}$, we seek to learn a model $h_\theta$ with maximal performance on unseen target data. Further, we focus on the task of $K$-way semantic segmentation, where for input image $x \in \mathbb{R}^{H \times W \times 3}$ we seek to make per-pixel predictions $\hat{y} \in \mathbb{N}^{H \times W \times K}$.

In Video-DAS, the input data is a series of $N_\mathcal{S}$ source videos and $N_\mathcal{T}$ target videos of length $\tau$ given by $D_\mathcal{S} = \{[(x_1, y_1), ..., (x_\tau, y_\tau)]_i \mid i = 1, ..., N_\mathcal{S}\}$ and $\mathcal{D}_\mathcal{T} = \{[x_1, ..., x_\tau]_i \mid i = 1, ..., N_\mathcal{T}\}$. As additional input, we can estimate the optical flow $o_{t \to t+k}$ between frames $x_t$ and $x_{t+k}$ using a trained flow prediction model. Pixels mapped together by optical flow represent the same object at different points in time, so they should intuitively be assigned the same class. In the absence of target labels, Video-DAS methods seek to leverage this temporal signal.

### 3.2 Overview of Video-DAS and Image-DAS

**Review of Video-DAS methods**

**1) DA-VSN (Guan et al., 2021).** Domain Adaptive Video Segmentation Network (DA-VSN) was one of the first Video-DAS approaches, relying on a standard image discriminator and an additional video-level discriminator which acts on stacked features from consecutive frames. Further, it optimizes unconfident predictions from frame $x_t$ to match confident predictions in the previous frame $x_{t-1}$.

**2) TPS (Xing et al., 2022).** Temporal Pseudo Supervision (TPS) improves upon DAVSN via pseudo-label based self-training, which jointly optimizes source cross entropy and temporal consistency losses. The temporal consistency loss warps predictions made for the previous frame forward in time (via optical flow), and uses the warped predictions as a pseudo-label for the augmented current frame.

**3) STPL (Lo et al., 2023).** STPL tackles the Video-DAS problem with an additional source-free constraint. Notably, they employ contrastive losses on spatio-temporal video features to obtain strong performance. However, we are unable to compare against their method as their code is not public.

**4) I2VDA (Wu et al., 2022).** I2VDA only leverages videos in the target domain, via temporal consistency regularization: predictions $h_\theta(x_t)$ and $h_\theta(x_{t+1})$ are merged in an intermediate space, which are then propagated to $t+1$ and supervised against the corresponding pseudo-label $\hat{y}_{t+1}$ via cross entropy loss.

**5) MOM (Cho et al., 2023).** Moving Object Mixing (MOM) sets the current SOTA in Video-DAS and relied primarily on a custom class-mix operation, which blends images from source and target domain. Concretely, they extend class-mix to videos with consistent mix-up which ensures that classmix is temporally consistent. Further they, employ consistency-based pseudo-label refinement, and image-level feature alignment between source and mixed domain images.

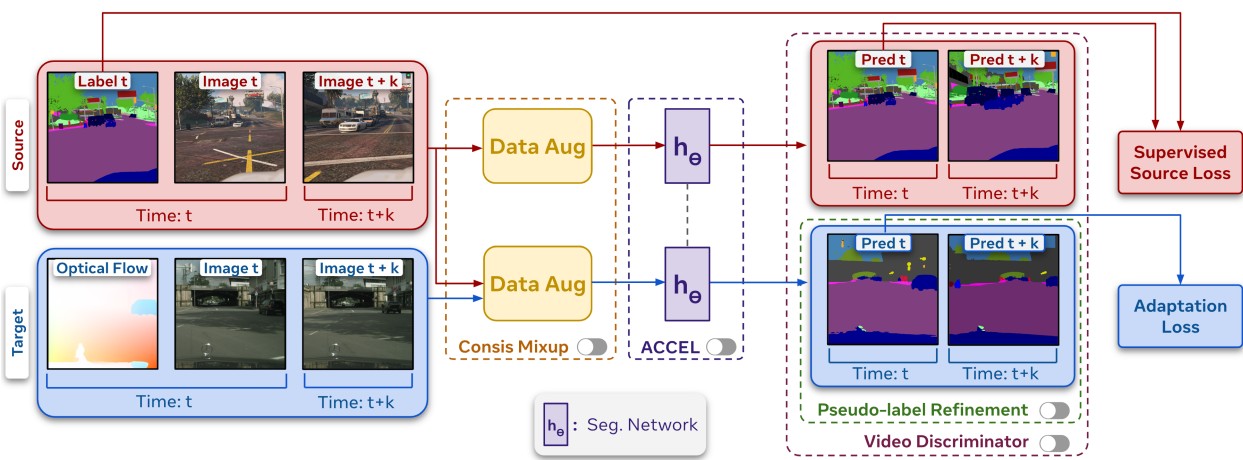

Figure 2: **Simplified Training Pipeline.** Temporally separated frames are first augmented and then passed through a model $h_\theta$ to produce source and target predictions, which produce supervised and adaptation losses. In this standard self-training pipeline, Video-DAS methods typically add one or more of the following techniques: consistent mixup, ACCEL, pseudo-label refinement, and video discriminators. We further elaborate on each of these techniques in Figure 3.

All of the works described above use a Resnet-101 (He et al., 2016) backbone with a DeepLabV2 (Chen et al., 2017) segmentation head, and most use the ACCEL (Jain et al., 2019) architecture. The ACCEL architecture is a wrapper, which separately generates segmentations for $x_t$ and $x_{t+k}$, propagates $x_{t+k} \rightarrow x_t$ via optical flow, and then fuses the two sets of predictions with a 1x1 convolutional layer. Table 1 provides a summary of the primary components used in prior Video-DAS works: video discriminators, architectural modifications (ACCEL), consistent mixup, and pseudo-label refinement. For more details on each approach see Appendix A.2.

**Image-DAS baseline.** For our updated baseline, we employ HRDA+MIC (Hoyer et al., 2022c), the culmination of a line of work (Hoyer et al., 2022a;b;c), which substantially improves the state-of-the-art in Image-DAS. We choose this line of work due to its substantive gains over previous Image-DAS methods, and note that numerous following Image-DAS works have also chosen to add on top of it (Chen et al., 2022a; Vayyat et al., 2022; Xie et al., 2023; Zhou et al., 2023; Ettedgui et al., 2022).

Concretely, HRDA+MIC combines i) student-teacher self-training with rare class sampling and an ImageNet feature-distance-based regularization objective, ii) Multi-Resolution Fusion (MRFusion) which enables models to train on full resolution images while maintaining a manageable memory footprint by fusing predictions from high and low resolution crops, and iii) an auxiliary masked image consistency (MIC) objective which generates segmentation masks from a masked version of an image. For details see Appendix A.1.

### 3.3 Overview of Video-DAS techniques

In Table 1, we summarized key techniques from prior Video-DAS works, specifically, video domain discriminators, the ACCEL architecture (Jain et al., 2019), temporally-consistent mix-up (Cho et al., 2023), and pseudo-label refinement. Now, we explain each of these techniques in depth (Fig. 3), as well as how they are integrated into our overall pipeline (Fig. 2).

**Preliminaries:** We first define the optical flow based propagation operation (prop) that both ACCEL and pseudo-label refinement rely on. Intuitively, $\text{prop}(x_{t+k}, o_{t \rightarrow t+k})$ aligns image $x_{t+k}$ with $x_t$ based on the optical flow between the two frames $o_{t \rightarrow t+k}$. Intuitively, $o_{t \rightarrow t+k}$ is a mapping from each pixel in $x_t$ to a unique pixel in $x_{t+k}$. Mathematically, for each pixel $[i, j]$ we have

$$\text{prop}(x_{t+k}, o_{t \rightarrow t+k})[i, j] = x_{t+k}[i + o_y, j + o_x]$$

where $o_x$ and $o_y$ are the $x$ and $y$ components of $o_{t \rightarrow t+k}$. For simplicity, we will denote this with $\text{prop}(x_{t+k})$.

(a) Consistent Mixup + Accel

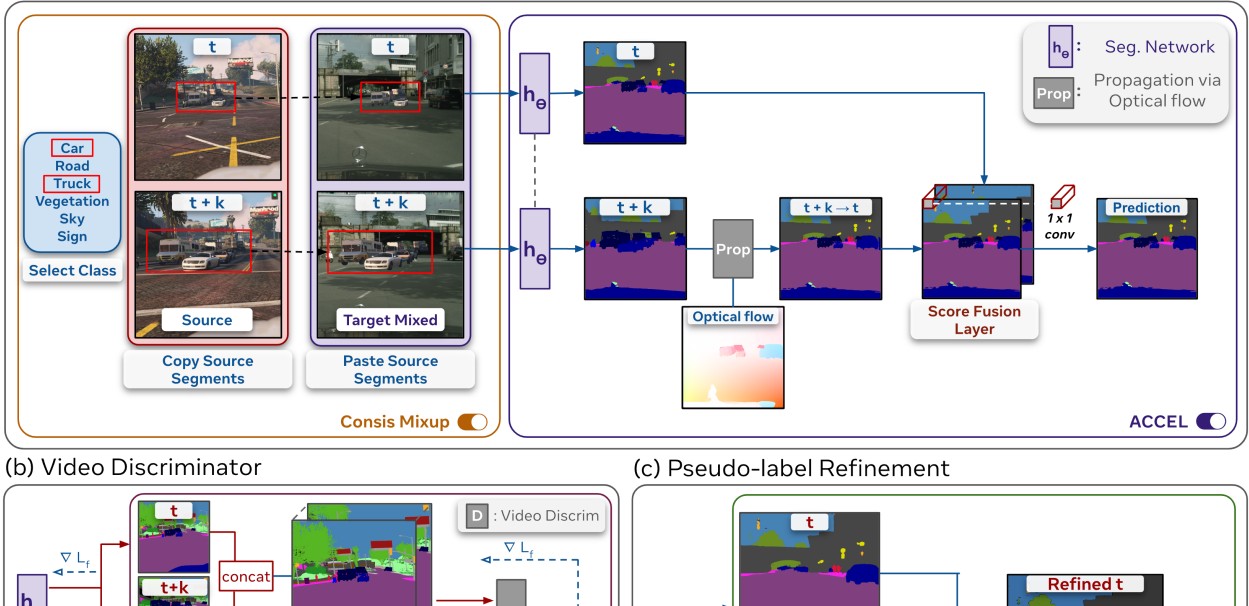

(b) Video Discriminator       (c) Pseudo-label Refinement

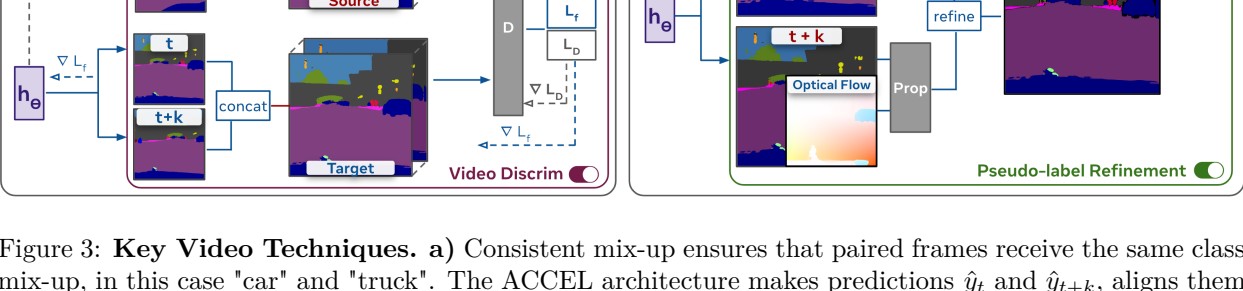

Figure 3: **Key Video Techniques. a)** Consistent mix-up ensures that paired frames receive the same class mix-up, in this case "car" and "truck". The ACCEL architecture makes predictions $\hat{y}_t$ and $\hat{y}_{t+k}$, aligns them via $\texttt{prop}(y_{t+k}, o_{t \to t+k})$, then fuses them with a $1 \times 1$ convolution. **b)** Video discriminators learn a classifier to distinguish whether temporally stacked features belong to the source or target domain, in conjunction with a feature encoder that is updated adversarially to make the two indistinguishable. **c)** Pseudo-label refinement improves predictions by fusing $\hat{y}_t$ and $\hat{y}_{t+k}$ based on one of several criteria.

▷ **Video discriminator:** Following (Tsai et al., 2018), we jointly train an auxiliary domain discriminator network $D$ (to classify whether intermediate features belong to a source or target sample), and a feature encoder to make source and target features indistinguishable. Following (Guan et al., 2021), we feed in *concatenated features* for temporally separated frames $P = h_\theta(x_t) \oplus h_\theta(x_{t+k})$, and optimize a domain adversarial loss (Tzeng et al., 2017):

$$\min_D \mathcal{L}_D = -\sum_{x_{\mathcal{S}} \in D_{\mathcal{S}}} \sum_{i,j} \log \mathbf{D}(P(x_{\mathcal{S}}))[i,j] - \sum_{x_{\mathcal{T}} \in D_{\mathcal{T}}} \sum_{i,j} \log(1 - \mathbf{D}(P(x_{\mathcal{T}})[i,j]);$$

$$\min_\theta \mathcal{L}_f = -\sum_{x_{\mathcal{T}} \in D_{\mathcal{T}}} \sum_{i,j} \log \mathbf{D}(P(x_{\mathcal{T}})[i,j])$$

▷ **ACCEL:** ACCEL (Jain et al., 2019) is a video segmentation architecture which fuses predictions for consecutive frames. Specifically, we compute individual frame logits $\hat{y}_t = h_\theta(x_t)$ and $\hat{y}_{t+k} = h'_\theta(x_{t+k})$. The logits are then aligned via $\hat{y}'_t = \texttt{prop}(\hat{y}_{t+k})$ and stacked along the channel dimension (H, W, 2C). This is fused into a unified prediction (H, W, C) via a 1x1 convolutional layer $L_{\text{ACCEL}}(x_t, x_{t+k}) = \text{conv}_{1 \times 1}(\hat{y}_t \oplus \hat{y}'_t)$. During training, gradients are only backpropagated through the $f(x_t)$ branch. Further, note that this architecture assumes access to both frames $x_t$ and $x_{t+k}$ at evaluation time. Following prior works in Video-DAS, we share weights across $h_\theta$ and $h'_\theta$.

(a) Consistency

(b) Max Confidence

(c) Warped Frame

(d) Oracle

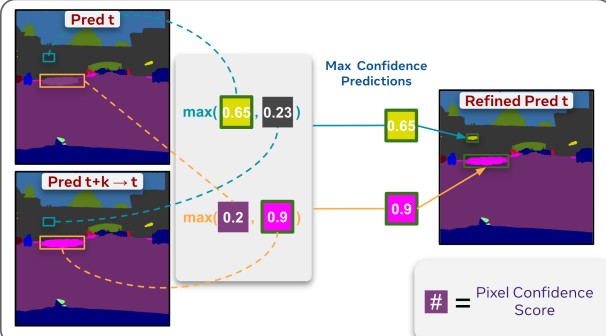
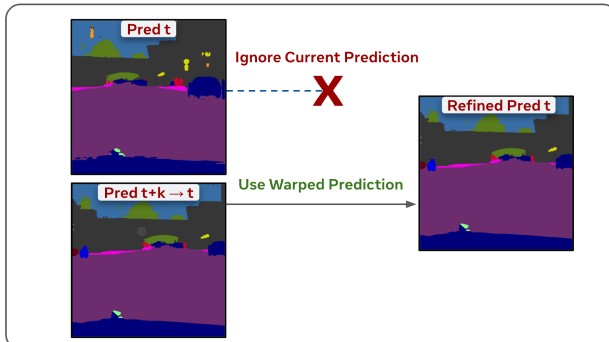
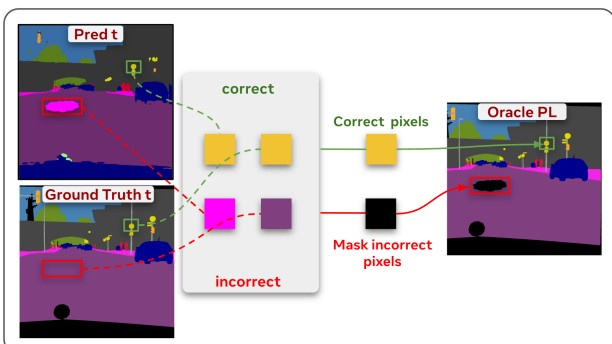

Figure 4: Each pseudo-label refinement strategy takes as input the current prediction $\hat{y}_t$, as well as a prediction for a future frame $\hat{y}_{t+k}$ warped via optical flow to the current frame, and merges them together to make a refined prediction $\hat{y}'_t$. **a)** Consistency discards predictions that are inconsistent across frames. **b)** Max confidence selects the more confident prediction between frames. **c)** Warped frame uses the warped prediction instead of the current prediction. **d)** Oracle performs consistency based refinement, but with the ground truth label.

▷ **Consistent Mixup:** Class-mixing has emerged as a popular component of several UDA methods. Specifically, the mix operation $\texttt{mix}(x_1, x_2, k)$ copies all pixels belonging to class $k$ in $x_1$ to the same position in $x_2$. Many UDA approaches utilize the class-mix augmentation across source and target images $\texttt{mix}(x_S, x_T, k)$, and correspondingly across source ground truth and target pseudo-labels $\texttt{mix}(y_S, \hat{y}_T, k)$. Further, in order to correctly apply the class-mix augmentation on models which take $x_t$ and $x_{t+k}$ as input (such as ACCEL), (Cho et al., 2023) propose *consistent* mix-up, wherein they pick the *same* set of classes for both frames $x_t$ $x_{t+k}$ so not to break optical flow correspondence.

▷ **Pseudo-label Refinement:** Several Video-DAS methods utilize the redundancy in information between two consecutive frames to refine pseudo-labels. Broadly speaking, these strategies take pseudo-labels for two timesteps $\hat{y}_t, \hat{y}_{t+k}$, along with their optical flow $o_{t \to t+k}$, and output an improved pseudo-label $\hat{y}'_t$. Below we list a few common strategies (see Figure 4 for an overview). For simplicity, we drop per-pixel indexing $y^{(i,j)}$:

**a) Consistency** simply discards any pixel which is inconsistent after optical flow propagation between $\hat{y}_t, \hat{y}_{t+k}$ and was used in (Cho et al., 2023), as follows: $\hat{y}'_t = \begin{cases} \hat{y}_t & \hat{y}_t = \texttt{prop}(\hat{y}_{t+k}) \\ \texttt{NAN} & \hat{y}_t \neq \texttt{prop}(\hat{y}_{t+k}) \end{cases}$

**b) Max confidence** selects the more confident prediction between $\hat{y}_t, \hat{y}_{t+k}$, similar to (Guan et al., 2021). Let $\hat{p}_t$ and $\hat{p}_{t+k}$ denote the corresponding top-1 confidence values, we set $\hat{y}'_t = \begin{cases} \hat{y}_t & \hat{p}_t < \texttt{prop}(\hat{p}_{t+k}) \\ \texttt{prop}(\hat{y}_{t+k}) & \text{else} \end{cases}$

**c) Warp frame** simply sets $\hat{y}'_t = \text{prop}(\hat{y}_{t+k})$, intuitively using a future or past frame's propagated prediction as a pseudo-label for the current frame. For $k = -1$, this strategy matches (Xing et al., 2022). In our experiments, we test both $k = 1$ (forward warp) and $k = -1$ (backward warp).

**d) Oracle** is an upper bound strategy for pseudo-label refinement, which filters based on consistency with the ground truth $y$, given by: $\hat{y}'_t = \begin{cases} \hat{y}_t & \hat{y}_t = y_t \\ \texttt{NAN} & \hat{y}_t \neq y_t \end{cases}$

## 4  Experiments

We now present our experimental setup. First, we compare the efficacy of Video-DAS methods with state-of-the-art Image-DAS methods (Sec. 3.2). Next, we explore hybrid UDA strategies that combine techniques from both lines of work (Sec. 4.3). To do so, we benchmark representative approaches for TPS (Xing et al., 2022), DAVSN (Guan et al., 2021), UDA-VSS (Shin et al., 2021) and MOM (Cho et al., 2023), as well as each key Video-DAS component individually. Finally, we explore pseudo-label refinement strategies across datasets, architectures, and frame distance (Sec. 4.5).

### 4.1  Experimental Setup

**Datasets.** Prior work in Video-DAS (Guan et al., 2021; Xing et al., 2022; Cho et al., 2023) studies adaptation on two popular benchmarks: Viper (Richter et al., 2016) to Cityscapes-Seq (Cordts et al., 2016) and Synthia-Seq (Ros et al., 2016) to Cityscapes-Seq.

▷ **Viper→Cityscapes-Seq**. Viper contains 13367 training clips and 4959 validation clips, each of length 10. Cityscapes-Seq contains 2975 training clips and 500 validation clips, each of length 30. Performance is measured over the 20th frame in each Cityscapes-Seq clip for which labels are available. Note that Viper is similar to the GTAV dataset (Richter et al., 2017) popularly used in the Image-DAS literature, as both are curated from the GTAV video game. However unlike the GTAV dataset, Viper contains *sequentially* labeled frames. Similarly, Cityscapes-Seq is a strict superset of the Cityscapes split used in Image-DAS works: while Cityscapes-Seq contains complete video clips from the original dataset and is largely unlabeled, Cityscapes only contains every 20th frame for which labels are available. Viper contains a total of 31 classes, of which 15 overlap with Cityscapes, so we calculate mIoU over these 15 classes.

▷ **Synthia-Seq→Cityscapes-Seq**. Synthia-Seq is a considerably smaller synthetic dataset, with a total of 850 *sequential* frames. To mitigate the small size of the dataset, we train on every frame, consistent with prior work. Synthia-Seq contains a total of 13 classes, and we calculate mIoU over the 11 classes shared with Cityscapes.

▷ **BDDVid**. We introduce support for a new target domain dataset derived from BDD10k (Yu et al., 2020), which to our knowledge has not been studied previously in the context of Video-DAS. BDD10k is a series of 10,000 driving images across a variety of conditions with semantic segmentation annotation. Of these 10,000 images, we identify 3,429 with valid corresponding video clips in the BDD100k dataset, making this subset suitable for Video-DAS. We refer to this subset as BDDVid. Next, we split these 3,429 images into 2,999 train samples and 430 evaluation samples. In BDD10k, the labeled frame is *generally* the 10th second in the 40-second clip, but not always. To mitigate this, we ultimately only evaluate images in BDD10k that perfectly correspond with the segmentation annotation, while at training time we use frames directly extracted from BDD100k video clips.

**Metrics.** Following prior work, we report mean intersection-over-union (mIoU) over the Cityscapes-Seq validation set (which is identical to the Cityscapes validation set) and over the BDDVid validation set. We employ two evaluation paradigms popular in the literature: the first predicts $y_t$ given a single frame $x_t$, while the second does so given two frames $(x_t, x_{t-k})$. The dual frame approach is employed by a subset of approaches that we describe in more detail in Sec. 3.3.

**Implementation Details.** To run these experiments, we make a number of contributions building off of the MMSegmentation (MMSeg) codebase (Contributors, 2020). We extend MMSeg to load consecutive frames

Table 2: Existing approaches in Video-DAS fail to leverage the recent advances in Image-DAS. Even Moving Object Mixing, the state of the art in Video-DAS, underperforms HRDA+MIC which is an image only method. All approaches use a DeepLabV2 architecture with a Resnet-101 backbone. Experiments that we ran are marked with a *.

| | Approach | Viper→CS-Seq | Synthia-Seq→CS-Seq |
|---|---|---|---|
| | Source* | 36.7 | 30.1 |
| Video-DAS | DAVSN (Guan et al., 2021) | 47.8 | 49.5 |
| | TPS (Xing et al., 2022) | 48.9 | 53.8 |
| | I2VDA (Wu et al., 2022) | 51.2 | 53.0 |
| | STPL (Lo et al., 2023) | 52.5 | 51.8 |
| | Moving Object Mixing (Cho et al., 2023) | 53.8 | 56.3 |
| Image-DAS | HRDA* (Hoyer et al., 2022b) | 65.5 | **76.1** |
| | HRDA + MIC* (Hoyer et al., 2022c) | **68.3** | 75.3 |
| | Target* (upper bound) | 83.0 | 84.9 |

Table 3: To understand why HRDA+MIC (Hoyer et al., 2022c) greatly outperforms many Video-DAS methods, we perform an ablation study. All experiments use a DeepLabV2 architecture with a Resnet-101 backbone.

| Approach | Viper → CS-Seq | Synthia-Seq → CS-Seq |
|---|---|---|
| Full approach (Hoyer et al., 2022c) | 68.3 | 75.3 |
| - MIC | $65.5_{-2.8}$ | $76.1_{+0.8}$ |
| - MRFusion | $53.3_{-15.0}$ | $67.3_{-8.0}$ |
| - Rare class sampling | $52.4_{-15.9}$ | $66.2_{-9.1}$ |
| - ImgNet feature distance reg. | $50.7_{-17.6}$ | $66.4_{-8.9}$ |

with their corresponding optical flows. The optical flows are generated by Flowformer (Huang et al., 2022) on Viper (Richter et al., 2017), Synthia-Seq (Ros et al., 2016), BDDVid, and Cityscapes-Seq (Cordts et al., 2016). In addition, we provide implementations of many Video-DAS techniques such as video discriminators, consistent mix-up, and various pseudo-label refinement strategies.

MMSeg enables researchers to easily swap out backbones, segmentation heads *etc.*, which will help keep Video-DAS techniques up to date in the future. Throughout the paper we deploy two base architectures following prior work: Resnet101+DLV2 (He et al., 2016; Chen et al., 2017) and Segformer (Xie et al., 2021). After running a learning rate sweep we find the parameters used in HRDA+MIC (Hoyer et al., 2022c) to work best and so retain the AdamW optimizer (Loshchilov & Hutter, 2019) with a learning rate of 6e-5 (encoder), 6e-4 (decoder), batch size of 2, 40k iterations of training, and a linear decay schedule with a warmup of 1500 iterations. For all experiments, we fix the random seed to 1.

### 4.2 How do Video-DAS methods compare to updated Image-DAS baselines?

Table 2 presents results. We find that:

▷ **SoTA Image-DAS methods outperform specialized Video-DAS methods.** Specifically, HRDA+MIC beats the state-of-the-art in Video-DAS (MOM) by 14.5 mIoU on Viper→Cityscapes-Seq and 19.0 mIoU on Synthia-Seq→Cityscapes-Seq, even after controlling for model architecture. Importantly, it does so without explicitly leveraging the temporal nature of the data, while the Video-DAS approaches explicitly exploit it. Further, the improvements of HRDA+MIC are most pronounced on rare classes like traffic light (+33.7 mIoU Viper, +46.3 mIoU Synthia-Seq), traffic sign (+25.8 mIoU Viper, +37.3 mIoU Synthia-Seq) and bicycle (+24.0 mIoU Viper). We present all class-wise IoUs in Tables 10 and 11. Note

Table 4: We test adding existing Video-DAS methods to our Image-DAS baseline (HRDA with DeepLabV2), as well as additional custom combinations of the key techniques used in prior works. We bold the best performing method and underline the second-best. Vid=Video Discriminator, ACCEL=ACCEL+Consistent Mixup, PL=Pseudo-label Refinement, Syn=Synthia-Seq, Vip=Viper and BDD=BDDVid

| Name | Vid | ACCEL | PL | Vip→CS | Syn→CS | Vip→BDD | Syn→BDD |
|------|-----|-------|-----|--------|--------|---------|---------|
| Source | | | None | 36.7 | 30.1 | 36.6 | 25.4 |
| HRDA (baseline) | | | None | 65.5 | 76.1 | 49.1 | 48.1 |
| + TPS | | ✓ | Warp Frame | 64.9 | 72.9 | 49.3 | 51.0 |
| + DAVSN | ✓ | ✓ | MaxConf | 65.3 | 75.2 | 49.7 | **52.1** |
| + UDA-VSS | ✓ | | Consis | 64.3 | 76.9 | 50.5 | 50.7 |
| + MOM | | ✓ | Consis | 65.9 | 75.2 | 47.6 | 51.8 |
| + Custom | ✓ | | None | 63.4 | 75.9 | **50.6** | 49.6 |
| + Custom | | ✓ | None | **66.9** | 76.6 | 50.0 | 50.3 |
| + Custom | | | Consis | 65.8 | 76.5 | 49.7 | 51.6 |
| + Custom | ✓ | ✓ | None | 63.7 | 76.1 | 48.9 | 49.7 |
| + Custom | | ✓ | Consis | 65.9 | 75.2 | 47.6 | 51.8 |
| + Custom | ✓ | | Consis | 64.3 | **76.9** | 50.5 | 50.7 |
| + Custom | ✓ | ✓ | Consis | 66.0 | 76.7 | 49.6 | 51.5 |
| Target (Upper Bound) | | | | 86.1 | 87.2 | 55.8 | 59.9 |

that in Table 2 which compares against prior work, we evaluate two established benchmarks, but in Table 4 we construct additional shifts to draw stronger conclusions.

▷ **Multi-resolution fusion (MRFusion) is the biggest contributor to improved performance.** To explain the strong gains obtained by HRDA+MIC, we perform an ablation study in Table 3 wherein we systematically strip away components from the full approach. A large portion of the performance gap is due to HRDA's MRFusion strategy (+12.2 mIoU Viper→Cityscapes-Seq, 8.8 mIoU Synthia→Cityscapes-Seq). This component allows HRDA to utilize full-resolution images whereas prior Video-DAS works are forced to down-sample their images to satisfy memory constraints. Additionally, the remaining components (MIC, RCS and ImageNet Feature Distance) each contribute a few points of performance improvement.

A natural question that arises from our analysis is whether we can merge the key techniques from Image-DAS and Video-DAS to further boost performance. We study this in the next subsection.

## 4.3 Can we combine techniques from Image-DAS and Video-DAS to improve performance?

We now evaluate whether the state-of-the-art image baseline (Hoyer et al., 2022c;b) can be further improved by incorporating the key techniques from Video-DAS. To reduce the computational burden, we use HRDA (rather than HRDA + MIC) as our starting point. Specifically, we try:

(a) Adding a video discriminator to the HRDA objective, in total optimizing $\lambda \mathcal{L}_{adv} + \mathcal{L}_{HRDA}$ (we find $\lambda = 0.1$ to work the best).

(b) Swapping out our architecture with ACCEL (Jain et al., 2019).

(c) Adding consistent cross-domain mix-up by ensuring that $x_t, x_{t+k}$ are given the same classmix augmentation. Note that consistent mixup is meaningful only when using a multi-frame input architecture such as ACCEL, and so we always use the two in conjunction.

(d) A modified version of HRDA that uses refined pseudo-labels for self-training, where refinement is performed using one of the strategies described above.

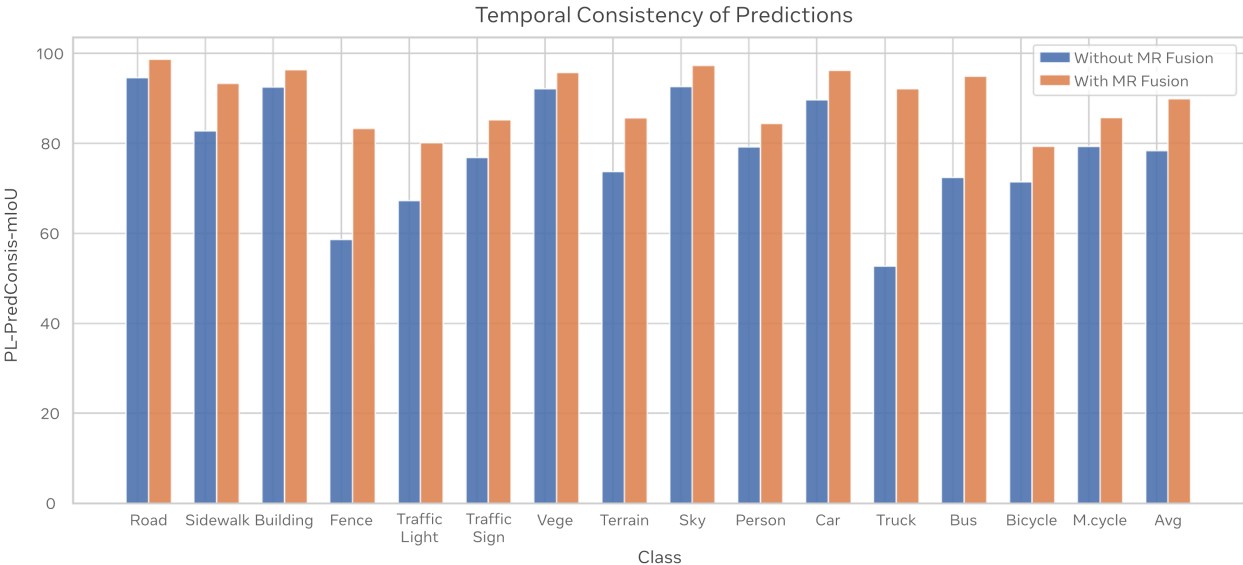

Figure 5: Temporal Consistency of Predictions (`PL-PredConsis-IoU`) with and without MRFusion. DeeplabV2 backbone trained on Viper→Cityscapes-Seq.

**Results.** In Table 4, we benchmark combinations of HRDA with prior Video-DAS works, in addition to other valid combinations of the common techniques identified in Video-DAS. Note that most prior Video-DAS methods can be constructed using a combination of one or more of the techniques mentioned above. These approaches are meant to be a naive way to update prior Video-DAS work to leverage the strong HRDA baseline. As seen, no technique outperforms the rest on all datasets. However, we do see some consistent gains across datasets by leveraging ACCEL (0.5-2.2 mIoU) and consistency based pseudo-label refinement (0.3-3.5 mIoU), with the largest improvements on the Synthia-Seq→BDDVid shift. While these initial improvements are promising, neither pseudo-label refinement nor ACCEL dominate across datasets. In addition, while at the upper bound we see an improvement of 3.5 mIoU, on many shifts this doesn't hold, and we see improvements of just 0.3 mIoU. And for perspective, all these gains are relatively small compared to the image level interventions observed in Table 3. Thus, we encourage future work to take these learnings to craft approaches with more significant improvements across multiple datasets.

## 4.4 Why is it difficult to combine Image-DAS and Video-DAS?

We now propose a *redundancy* hypothesis to explain why Video-DAS techniques do not further improve performance on top of state-of-the-art Image-DAS methods. We hypothesize that HRDA's multi-resolution fusion (MRFusion)-based self-training strategy leads to predictions that are not only highly accurate but also *highly temporally consistent*, obviating the need for further temporal refinement. To validate this, we first define a custom metric `PL-PredConsis`, which estimates the temporal predictive consistency of pseudolabels by computing mIoU between pseudolabels for the current frame $\hat{y}_t$ and warped pseudolabels for a future frame $\hat{y}'_t = \texttt{prop}(\hat{y}_{t+k})$, as:

$$\texttt{PL-PredConsis-mIoU}(\hat{y}_{t+k}, \hat{y}_t) = \texttt{mIoU}(\hat{y}'_t, \hat{y}_t)$$

In Figure 5, and Table 9 of the appendix, we report per-class and mean `PL-PredConsis` values for HRDA models trained without (top row) and with (bottom row) MRFusion – as seen, models trained with MRFusion exhibit *significantly* higher temporal consistency than those trained without (89.9 vs 78.4 on Viper→Cityscapes-Seq and 86.7 vs 83.3 on Synthia-Seq→Cityscapes-Seq). Recall that Video-DAS strategies typically seek to leverage temporal signals from adjacent frames to improve model predictions for the current frame – however, in the presence of MRFusion temporal consistency is already so high that there is

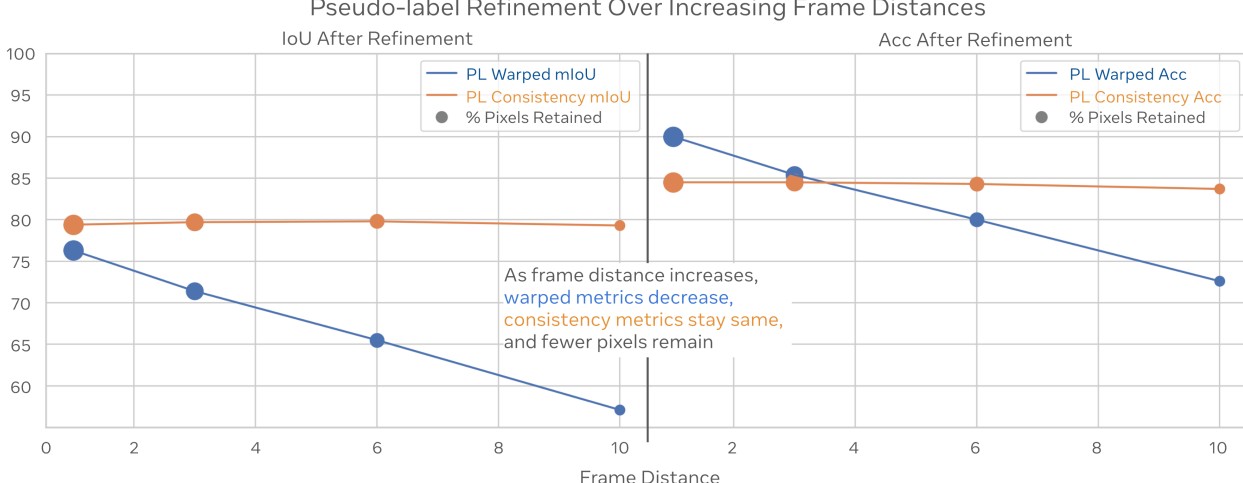

Figure 6: **Pseudo-label refinement over larger frame distances on Viper→Cityscapes-Seq.** At larger frame distances, we observe a drop in warped metrics and percentage of pixels retained, while consistency metrics remain stagnant. This suggests that pseudo-label refinement is worse over larger frame distances.

simply not much temporal signal to leverage. Further, MRFusion is designed to improve recognition of small objects by leveraging detail from high-resolution inputs – accordingly we find that that temporal consistency is greatly improved across these classes like 'fence', 'traffic sign', 'traffic light'.

### 4.5 Exploring pseudo-label refinement strategies

In light of the limited improvements offered by consistency-based pseudo-label refinement as found in the previous subsection, we now perform a comprehensive analysis of pseudo-label refinement strategies. Notably, we can perform analysis on stronger architectures like Segformer (Xie et al., 2021), because pseudo-label refinement is less memory intensive than ACCEL or video discriminators, which we benchmark in Table 4.

We report results across architecture in Tables 5-8 of the appendix. We find that:

▷ **Pseudo-label refinement is more effective on Synthia-Seq than Viper.** We find that on Synthia-Seq, pseudo-label refinement improves performance in 3/4 settings (0.7-2.4 mIoU improvement), in fact, improving upon the state-of-the-art using backward flow and max confidence (77.9 vs 76.6 mIoU, Table 5). On Viper, we observe less consistent improvements, with +4.6 mIoU on HRDA Segformer, and +0.3 mIoU on HRDA DeepLabV2. One possible explanation is that pseudo-label refinement is more effective when there is less labeled data as is the case with Synthia-Seq, but we leave further analysis to future work.

▷ **Oracle pseudo-label refinement lags behind fully supervised performance.** Across settings, we observe a considerable gap between the HRDA baseline and the "target only" oracle approach (on average +14.3 on Viper→Cityscapes-Seq and +11.0 on Synthia-Seq→Cityscapes-Seq). With oracle pseudo-label refinement, this gap shrinks considerably (on average +4.9 on Viper→Cityscapes-Seq and +2.7 on Synthia-Seq→Cityscapes-Seq), suggesting that while more precise pseudo-label refinement methods can improve performance, even perfect refinement is insufficient to match supervised performance.

▷ **Refining pseudo-labels via traditional methods across larger frame distances hurts.** Finally, we investigate whether refinement over larger frame distances produces stronger pseudo-labels. All prior Video-DAS works use frames $x_t, x_{t-1}, x_{t-2}$ for pseudo-label refinement, whereas we consider the impact of using $x_t, x_{t+k}$ for $k \in \{1, 3, 6, 10\}$. To investigate whether using larger frame distances is beneficial, we take a trained HRDA Segformer model and run evaluation with two new custom metrics, `PL-Warped` and `PL-Consistency`. `PL-Warped` takes a future prediction $\hat{y}_{t+k}$ and current ground truth $y_t$. We warp $\hat{y}_{t+k}$

onto the current frame to generate $\hat{y}'_t = \texttt{prop}(\hat{y}_{t+k})$. We then define:

$$\texttt{PL-Warped-mIoU}(\hat{y}_{t+k}, y_t) = \texttt{mIoU}(\hat{y}'_t, y_t) \qquad \texttt{PL-Warped-Acc}(\hat{y}_{t+k}, y_t) = \texttt{Acc}(\hat{y}'_t, y_t)$$

A `warped` metric measures whether future frames provide more accurate pseudo-labels than the current frame, assuming perfect propagation. This intuitively tests if getting closer to an object improves predictions on it. Note `Acc` is class-wise average accuracy. Similarly, a consistency metric helps us measure whether temporally consistent predictions are likely to be correct.

$$
\begin{aligned}
\texttt{PL-Consistency-mIoU}(\hat{y}_{t+k}, \hat{y}_t, y_t) &= \texttt{mIoU}((\hat{y}'_t == \hat{y}_t), y_t) \\
\texttt{PL-Consistency-Acc}(\hat{y}_{t+k}, \hat{y}_t, y_t) &= \texttt{Acc}((\hat{y}'_t == \hat{y}_t), y_t)
\end{aligned}
\tag{1}
$$

As shown in Figure 6 , we find that pseudo-label refinement becomes less effective over larger frame distances: `PL-Warped` quickly decreases with increasing frame distances, while `PL-Consistency` remains stagnant even as the proportion of pixels retained after filtering decreases.

## 5    Discussion and Future Work

The field of Video-DAS is currently in an odd place. Image-based approaches have advanced so much that they simply outperform video methods, largely because of techniques such as multi-resolution fusion. However, certain Video-DAS techniques still show occasional promise, such as ACCEL and pseudo-label refinement. Further, while we find that no single pseudo-label refinement strategy performs well across settings, such approaches are generally more effective on Synthia suggesting suitability to the the low data setting. However, our initial exploration leads us to believe that static, hand-crafted refinement strategies based on heuristics are likely too brittle to consistently improve performance, and adaptive, learning-based, approaches (like ACCEL) perhaps require more consideration. Further, while we show Image-DAS methods to be far superior to Video-DAS methods for sim-to-real semantic segmentation adaptation, we leave to future work an investigation into whether this holds more generally. Perhaps Video-DAS will only lead to consistently significant improvements for segmentation on harder domain shifts (such as BDDVid). Further, in tasks like action recognition which necessitate a model to understand a series of frames, video may be necessary.

To accelerate research in Video-DAS, we open source our code built off of MMSegmentation, and provide multi-frame support I/O, optical flow, new domain shifts, and baseline implementations of key Video-DAS methods. We believe that the lack of a standardized codebase for Video-DAS is one of the primary reasons for the siloed progress in the field, and hope that with our codebase future work can both easily cross-benchmark methods and develop specialized Video-DAS methods built on top of the latest and greatest in image-level semantic segmentation.

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

# A Appendix

## A.1 Image-DAS Methods

HRDA+MIC falls under the self training style of UDA. The core training objectives are to minimize cross entropy loss on source domain images with ground truth labels, as well as cross entropy on target domain images with pseudo-labels. At each iteration pseudo-labels are generated via a teacher network $\phi$ which is defined as an EMA update of the student $\theta$ following (Tarvainen & Valpola, 2018).

HRDA introduced Multi Resolution Fusion (MRFusion), which fuses high res and low res image crops, allowing the model to use high resolution details with a manageable GPU footprint. The context and detail predictions $\hat{y}_c, \hat{y}_d$ are resized $\zeta$ with scale $s$, then fused with a learnable attention module $a$.

$$\hat{y} = \zeta \left((1-a) \odot \hat{y}_c, s\right) + \zeta \left(a, s\right) \odot \hat{y}_d.$$

MIC additionally introduced a masked reconstruction objective, which attempts to predict segmentation maps from *masked* target images $x^M$. Given pseudo-label $p^T = \phi(x)$, pseudo-weight $q$, and masked prediction $\hat{y}^M = \phi(x^M)$

$$L_{mic} = qL_{ce}(\hat{y}^M, p^T).$$

ImageNet feature distance regularizes source domain predictions based on ImageNet (Deng et al., 2009) representations. Finally, Rare Class Sampling encourages drawing source domain examples containing rare classes. A class $c$ is drawn with frequency $P(c)$ where $f_c$ is the proportion of pixels with label $c$

$$P(c) = \frac{e^{(1-f_c)/T}}{\sum_{c'=1}^{C} e^{(1-f_{c'})/T}}.$$

## A.2 Video-DAS Methods

An early work in Video-DAS presents DAVSN (Guan et al., 2021), which broadly falls under the domain discriminative style of UDA. Their approach has an image discriminator, video discriminator, and a divergence loss between these discriminators. The image discriminator simply tries to distinguish source vs target features. Similarly the video discriminator tries to distinguish source video features and target video features. These video features are simply stacked features from two consecutive frames. The divergence loss helps each discriminator focus on different information. In addition to these losses, DAVSN explicitly optimizes for unconfident predictions in the current frame to match confident predictions from the previous frame $\mathcal{L}_{itcr}(G) = S\left(E\left(p_k^{\mathbb{T}}\right) - E\left(\hat{p}_{k-1}^{\mathbb{T}}\right)\right)\left|p_k^{\mathbb{T}} - \hat{p}_{k-1}^{\mathbb{T}}\right|$.

Another early work in Video-DAS presents UDA-VSS (Shin et al., 2021), an approach which relies on a standard video discriminator, and self training algorithm. The self training algorithm notably aggregates predictions from the future and previous frames, to improve pseudo-label generation. And these pseudo-labels are further refined by ignoring pseudo-labels predictions which are inconsistent between neighboring frames, via optical flow alignment. Note UDA-VSS reports on Viper to Cityscapes-VPS and Synthia-Seq to Cityscapes-VPS, which is slightly different that Cityscapes-Seq. Thus we cannot directly compare to the reported numbers in Table 2.

DAVSN is improved upon in TPS (Xing et al., 2022) by adopting pseudo-label based self training. There are only two losses, source cross entropy, and temporal consistency. The temporal consistency loss simply uses the previous frame warped forward in time via optical flow as a pseudo-label for the augmented current frame.

STPL (Lo et al., 2023) tackles the Video-DAS problem, but operates in the source free DA setting. Notably they employ contrastive losses on spatio-temporal video features. However, they do not have code available, so we do not benchmark their approach.

I2VDA (Wu et al., 2022) tackles the same Video-DAS problem, but designs an approach which only leverages videos in the target domain. Their key technique is temporal consistency regularization. First predictions

Table 5: A comparison of pseudo-label refinement strategies with Segformer backbone and HRDA + MIC. We bold the best refinement strategy and underline the second best. Refinement helps on Synthia-Seq but not Viper.

| Name | Flow Direction | PL Refine | Viper→CSSeq | Synthia-Seq→CSSeq |
|---|---|---|---|---|
| Source | | | 46.1 | 52.0 |
| HRDA+MIC | | | 76.4 | 76.6 |
| Custom | Backward | warp frame | 74.4 | 75.5 |
| Custom | Backward | consis | 75.0 | 76.0 |
| Custom | Backward | maxconf | 74.0 | **77.9** |
| Custom | Forward | warp frame | 73.2 | 72.6 |
| Custom | Forward | consis | 74.0 | 73.6 |
| Custom | Forward | maxconf | **75.3** | 74.2 |
| Oracle | | | 80.0 | 84.2 |
| Target | | | 86.1 | 87.2 |

Table 6: A comparison of pseudo-label refinement strategies with Segformer backbone and HRDA (No MIC). We bold the best refinement strategy and underline the second best. Refinement improves on both Viper and Synthia-Seq benchmarks.

| Name | Flow Direction | PL Refine | Viper→CSSeq | Synthia-Seq→CSSeq |
|---|---|---|---|---|
| Source | | | 46.1 | 52.0 |
| HRDA | | | 70.7 | 72.4 |
| Custom | Backward | warp frame | 71.5 | 69.0 |
| Custom | Backward | consis | 74.8 | **74.8** |
| Custom | Backward | maxconf | 70.7 | 74.4 |
| Custom | Forward | warp frame | 74.3 | 69.4 |
| Custom | Forward | consis | **75.3** | 74.5 |
| Custom | Forward | maxconf | 73.9 | 73.2 |
| Oracle | | | 82.3 | 83.8 |
| Target | | | 86.1 | 87.2 |

$f(x_t)$ and $f(x_{t+1})$ are merged in an intermediate space, then they are propagated to $t + 1$ and compared against the corresponding pseudo-label $\hat{y}_{t+1}$.

The current state of the art is set by Moving Object Mixing (MOM) (Cho et al., 2023), which differentiates itself via consistent mixup and a contrastive objective. Classmix is an augmentation in which pixels from a source image $x_S$ with labels in $c_{\text{classmix}}$ are pasted onto a target image $x_t$. Consistent mixup simply ensures that the same $c_{\text{classmix}}$ is chosen for both $x_t$ and $x_{t+k}$, where $k$ is the frame distance. The contrastive objective pulls together representations of a given class across source and mixed domain images. Given a mixed domain image, MOM filters out pixels which are inconsistent between two frames using optical flow. Then a feature centroid $f_c$ is calculated for each instance in the mixed image. Similarly, in the source domain a bank of feature centroids is stored for each class. MOM then performs L1 loss between $f_c$, and the most similar feature in the bank of source domain features for class $c$.

### A.3 Pseudo-label Refinement Results

We explore pseudo-label refinement across a number of dimensions: architecture, dataset, flow direction and refinement strategy. Tables 5, 6, 7 and 8 correspond to different architectures: HRDA+MIC on Segformer, HRDA on Segformer, HRDA+MIC on DLV2, HRDA on DLv2.

Table 7: A comparison of pseudo-label refinement strategies with DeepLabV2 backbone and MIC. We bold the best refinement strategy and underline the second best. In this case refinement is worse than the baseline.

| Name | Flow Direction | PL Refine | Viper→CSSeq | Synthia-Seq→CSSeq |
|---|---|---|---|---|
| Source | | | 36.7 | 30.1 |
| HRDA MIC | | | 68.3 | 75.3 |
| Custom | Backward | warp frame | 62.9 | 71.1 |
| Custom | Backward | consis | 67.7 | **74.7** |
| Custom | Backward | maxconf | 65.0 | 73.8 |
| Custom | Forward | warp frame | 67.2 | 72.3 |
| Custom | Forward | consis | 66.8 | 74.5 |
| Custom | Forward | maxconf | **68.2** | 71.9 |
| Oracle | | | 78.9 | 82.8 |
| Target | | | 83.0 | 84.9 |

Table 8: A comparison of pseudo-label refinement strategies with DeepLabV2 backbone and HRDA (no MIC). We bold the best refinement strategy and underline the second best. Refinement helps slightly on both benchmarks.

| Name | Flow Direction | PL Refine | Viper→CSSeq | Synthia-Seq→CSSeq |
|---|---|---|---|---|
| Source | | | 36.7 | 30.1 |
| HRDA | | | 65.5 | 76.1 |
| Custom | Backward | warp frame | 62.9 | 73.8 |
| Custom | Backward | consis | **65.8** | **76.8** |
| Custom | Backward | maxconf | 63.9 | 75.7 |
| Custom | Forward | warp frame | 63.7 | 71.7 |
| Custom | Forward | consis | 64.5 | 74.9 |
| Custom | Forward | maxconf | 64.1 | 73.9 |
| Oracle | | | 77.3 | 82.7 |
| Target | | | 83.0 | 84.9 |

### A.4 Temporal-Consistency Results

Here we include the temporal consistency results on the Synthia-Seq→Cityscapes-Seq shift as well

Table 9: Temporal Consistency of Predictions (`PL-PredConsis-IoU`) with and without MRFusion. DeeplabV2 backbone trained on Synthia-Seq→Cityscapes-Seq.

| Method | road | side | buil | pole | light | sign | veg | sky | per | rider | car | PL-PredConsis-mIoU |
|---|---|---|---|---|---|---|---|---|---|---|---|---|
| No MRFusion | 87.6 | 83.0 | 88.3 | 66.1 | 72.4 | 77.3 | 92.6 | 94.4 | 79.6 | 81.3 | 93.6 | 83.3 |
| HRDA | 96.4 | 88.1 | 94.0 | 67.6 | 75.3 | 81.2 | 95.0 | 96.7 | 81.0 | 83.7 | 95.0 | 86.7 |

### A.5 Class-wise IoUs

In Tables 10 and 11 we provide the classwise IoUs for the HRDA+MIC baseline and each prior work in Video-DAS, controlling for architecture.

Table 10: Classwise IoUs on Viper→CSSeq. All methods use Resnet101+DLV2.

| Method | road | side | buil | fen | light | sign | veg | ter | sky | per | car | truck | bus | mot | bic | mIoU |
|---|---|---|---|---|---|---|---|---|---|---|---|---|---|---|---|---|
| HRDA+MIC | **95.5** | **71.5** | **91.2** | 21.1 | **66.2** | **73.1** | **89.8** | **47.3** | **92.3** | **79.6** | **89.7** | **48.6** | 54.9 | 38.5 | **64.6** | **68.3** |
| MOM | 89.0 | 53.8 | 86.8 | 31.0 | 32.5 | 47.3 | 85.6 | 25.1 | 80.4 | 65.1 | 79.3 | 21.6 | 43.4 | 25.7 | 40.6 | 53.8 |
| STPL | 83.1 | 38.9 | 81.9 | **48.7** | 32.7 | 37.3 | 84.4 | 23.1 | 64.4 | 62.0 | 82.1 | 20.0 | **76.4** | **40.4** | 12.8 | 52.5 |
| I2VDA | 84.8 | 36.1 | 84.0 | 28.0 | 36.5 | 36.0 | 85.9 | 32.5 | 74.0 | 63.2 | 81.9 | 33.0 | 51.8 | 39.9 | 0.1 | 51.2 |
| TPS | 82.4 | 36.9 | 79.5 | 9.0 | 26.3 | 29.4 | 78.5 | 28.2 | 81.8 | 61.2 | 80.2 | 39.8 | 40.3 | 28.5 | 31.7 | 48.9 |
| DAVSN | 86.8 | 36.7 | 83.5 | 22.9 | 30.2 | 27.7 | 83.6 | 26.7 | 80.3 | 60.0 | 79.1 | 20.3 | 47.2 | 21.2 | 11.4 | 47.8 |

Table 11: Classwise IoUs on Synthia-Seq→CSSeq. All methods use Resnet101+DLV2.

| Method | road | side | buil | pole | light | sign | veg | sky | per | rider | car | mIoU |
|---|---|---|---|---|---|---|---|---|---|---|---|---|
| HRDA+MIC | **94.5** | **65.6** | **91.4** | **49.7** | **62.6** | **66.9** | **90.2** | **91.8** | **78.2** | **45.8** | **91.2** | **75.3** |
| MOM | 90.4 | 39.2 | 82.3 | 30.2 | 16.3 | 29.6 | 83.2 | 84.9 | 59.3 | 19.7 | 84.3 | 56.3 |
| STPL | 87.6 | 42.5 | 74.6 | 27.7 | 18.5 | 35.9 | 69.0 | 55.5 | 54.5 | 17.5 | 85.9 | 51.8 |
| I2VDA | 89.9 | 40.5 | 77.6 | 27.3 | 18.7 | 23.6 | 76.1 | 76.3 | 48.5 | 22.4 | 82.1 | 53.0 |
| TPS | 91.2 | 53.7 | 74.9 | 24.6 | 17.9 | 39.3 | 68.1 | 59.7 | 57.2 | 20.3 | 84.5 | 53.8 |
| DAVSN | 89.4 | 31.0 | 77.4 | 26.1 | 9.1 | 20.4 | 75.4 | 74.6 | 42.9 | 16.1 | 82.4 | 49.5 |

## A.6 Impact of Multiple Frames on Image-DAS

In order to compare fairly between an Image-DAS method which uses only one frame per clip, and a Video-DAS method which uses multiple, we should allow the Image-DAS method to access the additional frames as well. Interestingly in Table A.6 we find that the Image-DAS methods actually perform worse when we naively add these additional frames to the training set. We anticipate this is because of the strong redundancy between frames. Regardless, throughout the paper we restrict our Image-DAS methods to only using a single frame since it performs better.

Table 12: We evaluated the effect of including additional unlabelled frames as part of the target dataset, to fairly compare MIC to approaches which utilize both $x_t$ and $x_{t+k}$. We find that simply including $x_t, I_{x+k}$ in our target dataset actually hurts performance.

| Approach | Target Frames | mIoU |
|---|---|---|
| HRDA DLV2 | $t$ only | 65.5 |
| HRDA DLV2 | $t, t+1$ | 64.4 |
| HRDA+MIC DLV2 | $t$ only | 68.3 |
| HRDA+MIC DLV2 | $t, t+1$ | 65.0 |

