# OpenReview forum: "We're Not Using Videos Effectively: An Updated Domain Adaptive Video Segmentation Baseline"
_TMLR — Accepted by TMLR_

### Review · Reviewer_RrVf · 2023-10-04

**Summary Of Contributions:**

This paper comprehensively examines the impact of cutting-edge image-based domain adaptation techniques on the domain adaptive video semantic segmentation task, thereby establishing a stronger baseline. Additionally, it delves into the applicability of widely used video-based domain adaptive techniques for this task and conveys insights about the necessary components. Furthermore, the authors have made the codebase open-source to facilitate further research in this area.

**Audience:**

Yes

**Claims And Evidence:**

Yes

**Requested Changes:**

Please refer to the "Strengths And Weaknesses" part.

**Strengths And Weaknesses:**

**Strengths:**

1: The ablation studies of advanced image-based and popular video-based techniques for domain adaptive video semantic segmentation are comprehensive and convincing.

2: The constructed strong baseline is solid and conveys insights into the necessary techniques and strategies for the domain adaptive video semantic segmentation.

3: The whole work is easy to understand and follow up on. The open-source codebase helps other researchers reproduce the results and develop their methods.

**Weaknesses:**

1: The usage of **"video domain adaptation"** and **"domain adaptive video semantic segmentation"**. To the best of my knowledge, the "domain adaptation" is generally specific to the classification task (both image-based and video-based [1] areas) or serves as a collective term, including many domain adaptation tasks. However, this paper only considers the "(domain adaptive) video semantic segmentation" task as mentioned in previous literature [2, 3]. Therefore, the authors are encouraged to rethink using the sentence "video domain adaptation" in the whole main text.

2: Is the setting of **batch size** correct? In the Implementation Details section, the authors say the batch size is 2. I want to know what the "2" means. Does it mean the total batch size among all GPUs or the number for each GPU?

3: The **"Preliminaries"** and **"Experiments"** sections are encouraged to be **re-organized**. First, the "Preliminaries" section generally introduces the background knowledge for the task, and that's what Section 3.1 does. But Section 3.2 introduces the "Experimental Setup," which generally belongs to a part of "Experiments." Similarly, Section 4.2 mainly discusses the domain adaptation techniques, while Section 4 is called "Experiments." I think it would be better if this paper consolidates the method part and merges Section 3.2 into the experiment part.

4: Verifying the proposed simplified baseline on other benchmarks. The developed strong baseline shows promising results. However, the authors only test the methods on two scenarios. Following MIC [4], testifying the model's effectiveness on DarkZurich and ACDC can make the results more comprehensive.


[1] Chen, Min-Hung, et al. "Temporal attentive alignment for large-scale video domain adaptation." ICCV, 2019.

[2] Hu, Ping, et al. "Temporally distributed networks for fast video semantic segmentation." CVPR, 2020.

[3] Guan, Dayan, et al. "Domain adaptive video segmentation via temporal consistency regularization." CVPR, 2021.

[4] Hoyer, Lukas, et al. "MIC: Masked image consistency for context-enhanced domain adaptation." CVPR. 2023.

---

> ### Author Response · Authors · 2023-11-08
> **Response**
>
> Thank you for the thorough review! We especially appreciate hearing that you think our work is easy to understand and follow up on! We respond to specific concerns below:
>
> >The usage of ‘video domain adaptation’ and ‘domain adaptive video semantic segmentation’”.
>
> As recommended, we have revised our terminology in the revised draft.  We have replaced all instances of VideoDA with Video-DAS, meaning Video Domain Adaptation for Semantic Segmentation.
>
> > Is the setting of batch size correct?
>
> Yes. Following MIC, we only train on a single GPU.  Thus, “2” represents the true global batch size.
>
> > The "Preliminaries" and "Experiments" sections are encouraged to be re-organized
>
> Thank you for this suggestion! As recommended, we have comprehensively reorganized these sections.
>
> > Verifying the proposed simplified baseline on other benchmarks.
>
> Great suggestion!  In our updated draft, we have included additional experiments on Viper->BDD-Vid (labeled subset of the Berkeley Deep Drive dataset for which original videos are available) and Synthia->BDDVid (please see general response for details).  We also considered ACDC and Dark Zurich as suggested by the reviewer, but the change between consecutive frames is too large to be suitable for video semantic segmentation.

---

### Review · Reviewer_Ep2P · 2023-10-24

**Summary Of Contributions:**

This paper investigates domain adaptation (DA) for video (semantic) segmentation task. Authors start from the assumption that video-DA methods use a set of benchmarks different from those used by image-DA methods, and realized that the latter are much better performing on video benchmarks as compared to the former techniques supposedly designed for such benchmarks and leveraging temporal information. In addition, it is also reported that combinations of image-DA and video-DA methods do not even lead to consistent performance improvements.
Performances are reported for essentially 2 use cases (Viper to Cityscapes and Synthia to Cityscapes), and show that the most performing image-DA method (HRDA+MIC, CVPR 2023) outperforms state-of-the-art video-DA algorithms by a large margin (+14.5 and +19.0) in mIoU (mean Intersection over Union).

**Audience:**

Yes

**Broader Impact Concerns:**

I don't see ethical concerns for this paper.

**Claims And Evidence:**

No

**Requested Changes:**

I would like to see such work revised in the following way.

Provide experiments with more datasets or more combinations of the considered datasets.

Ablation analysis should be better detailed, addressing why image-DA method is working better and what is missing in the video-DA methods for not reaching similar performance.

Explore more thoroughly the design of a method combining image-DA and video-DA mechanisms, or better, explore how to integrate in video-DA methods some strategies of image-DA techniques in order to assess whether (video-DA) performance can improve.

Possibly addressing a larger set of DA tasks, not only related to segmentation.

**Strengths And Weaknesses:**

I have found this work interesting from the perspective of providing the community a study on image-DA vs. video-DA techniques on tasks of common interest, which is the case of segmentation, but also of classification, and of action recognition, even if for the latter some major revision of the image-DA methods should have been carried out.
Good points of the paper regard the comparison of several video-DA methods for segmentation task, the comparisons with essentially a recent image-DA method (and its variant), and an ablation analysis. It is also addressed the hybrid case in which image-DA and video-DA algorithms are integrated, showing not so consistent better results.
As a whole, the experimental analysis is fine even if it is not always comfortable to follow the work given the large number of methods considered, most of the times shortly described. For example, HRDA and HRDA+MIC are not easy to catch from only the description reported in the paper (including the one in Appendix).

But, my main concern is related to significance of the work so organized.
In particular, first, I don't deem so interesting for the community the findings claimed in the paper unless the reasons why image-DA methods are working better for videos with respect to video-DA techniques are provided. Claiming that the MRFusion and MIC are the major responsible for the performance improvement is not sufficient, it'd be interesting to figure out why video-DA methods are not exploiting similar mechanisms, or how the latter can be integrated in video-DA methods for boosting performance. In other words, one would like to see why temporal information is not so relevant for the task at hand.
Moreover, using only one image-DA method is also a limiting factor: HRDA is a specific method, but there a number of other techniques that can be compared, which are not following the same mechanisms of HRDA and that may deserve some comparisons.

Second, just 2 types of experiments having Cityscapes as target is too limiting to me. Probably, the literature addressed only these 2 cases, but drawing conclusions from these experiments only is insufficient to my opinion. Maybe one can provide all possible combinations of the 3 datasets as source/target.

Third, as also quoted above, it would be more interesting to provide a comparison on image- vs. video-DA methods for an extended set of tasks, not only for segmentation, but I understand that this would likely lead to another paper. Indeed, it'd be interesting to figure out the differences and similarities of exploiting temporal/multiple frames information in a DA setting for a set of tasks.

Fourth, when combining image- and video-DA methods, several strategies and parameters tuning have been applied, and it is unclear whether these variations affects (and how much) the final performance. It is also strange that results on the methods' combination are presented only for Viper to Cityscapes and not for Synthia to Cityscapes (Table 4).

---

> ### Author Response · Authors · 2023-11-08
> **Response**
>
> Thank you for the thorough review! We address specific concerns below:
>
> >I don't deem so interesting for the community the findings claimed in the paper unless the reasons why image-DA methods are working better for videos with respect to video-DA techniques are provided.
>
> Completely agree! In our general response above, we have conducted an in-depth analysis of possible reasons for this, concluding that state-of-the-art ImageDA methods trained with multi-resolution fusion already have high temporal consistency, which renders temporal pseudolabel refinement algorithms commonly used for VideoDA redundant.
>
> > Moreover, using only one image-DA method is also a limiting factor.
>
> This is a valid concern! Please see our general response above for a detailed discussion.
>
> > Second, just 2 types of experiments having Cityscapes as target is too limiting to me.
>
> Thank you for pointing this out. After evaluating dataset options we found the Berkeley Deep Drive (BDD) dataset to be best suited for our task. Accordingly, we have conducted experiments on two additional shifts: Viper->BDDVid (labeled subset of the Berkeley Deep Drive dataset for which original videos are available) and Synthia->BDDVid, and include these results in our revised draft. Please see our general response above for details.
>
> > Third, as also quoted above, it would be more interesting to provide a comparison on image- vs. video-DA methods for an extended set of tasks, not only for segmentation, but I understand that this would likely lead to another paper.
>
> We absolutely agree that this is a valuable direction, and hope that future work follows up on this.  Thank you for understanding that this would likely constitute a second paper.
>
> > Fourth, when combining image and video-DA methods, several strategies and parameters tuning have been applied, and it is unclear whether these variations affects (and how much) the final performance.
>
> We seek some clarification regarding this point. When combining image and video DA methods (Table 4), we include results for an exhaustive set of combinations of popular video DA strategies combined with a state-of-the-art ImageDAS method. We assume that by parameters the reviewer is referring to method hyperparameters – the only hyperparameter we manually tune via a sweep is the loss weight for the  the video discriminator (section 4.3, Part a). All other hyperparameters are kept fixed across experiments, and we include a detailed list in Section 4.1.
>
> > It is also strange that results on the methods' combination are presented only for Viper to Cityscapes and not for Synthia to Cityscapes (Table 4).
>
> As recommended, we have updated Table 4 to include additional results for Synthia to Cityscapes.

---

### Review · Reviewer_P6Qq · 2023-10-25

**Summary Of Contributions:**

The paper studies the effectiveness of recent video domain adaptation (Video-DA) techniques beyond that of the state-of-the-art image domain adaptation (Image-DA) methods applied to video frames, i.e., without acknowledging any temporal and continuity aspect of video frames. The question is interesting and rightly-posed since video recognition, despite the theoretical promise, has then and again proved challenging to benefit from aspects of video beyond frame-based recognitions.

The paper focuses on semantic segmentation as the recognition task, synthetic-to-real as the domain adaptation task, with synthetic datasets as the source data (GTA-based) and real-world Cityscapes as the target data.

On the Image-DA side, it takes a recent thread of works which benefits from high resolution images and in several recent works has clearly achieved state-of-the-art performance. For the video-DA side it takes several recent works published in 2022 and 2023 computer vision venues.

The main take-away is that all the several recent Video-DA works are largely outperformed by the state-of-the-art Image-DA method on the selected tasks.

The paper then moves on to ablate which Image-DA component contributes most the performance which is found to be, by far, the high-resolution processing.

Finally, the paper studies whether any of the techniques used in the several Video-DA methods can bring additional improvement to the state-of-the-art Image-DA applied on video DA but is found to not be the case; at least neither consistently nor significantly despite occasional small improvements.

**Audience:**

Yes

**Claims And Evidence:**

No

**Requested Changes:**

- The statements throughout the paper (possibly including the title) should be accordingly delimited so it doesn’t read as if it is the general video domain adaptation that is the subject of the study,or the scope of conclusion statements corroborated by the empirical results. (important for claims and evidence to match)

- A discussion regarding the reasons behind the Video-DA techniques not improving over the state-of-the-art Image-DA method.

The paper has clear strengths, the other mentioned negative aspects do not fall within the criteria of TMLR but could be good to address regardless.

**Strengths And Weaknesses:**

**Strengths**

- Although video recognition has been shown before to be challenging to benefit from video-specific signals significantly beyond still-image signals of the individual frames, the margin in the results are indeed surprising and regardless this can be quite informative for the development of the important subfields of both machine learning (domain adaptation of temporal data) and computer vision (video semantic segmentation)
- there is an open-source code released for the benchmarking of Video-DA on image and video semantic segmentation using the implementation of the various components of the state-of-the-art techniques.
- several recurring components of Image-DA are ablated to find the most effective part of the state-of-the-art Image-DA for Video-DA with clear take-away regarding the high-resolution processing being a key factor.
- studies the effectiveness of adding video-specific components to state-of-the-art Image-DA.

**Weaknesses**
- there is only one line of work for Image-DA that is tested on videos. Those line of works already clearly outperform the prior methods on Image-DA by a relatively large margin which could, to some extent, explain the observation. For various reasons including better understanding of with what method the temporal information might not be as useful it would be good to include other image-DA baselines and/or components.
- the benchmark test-bed is quite limited: only semantic segmentation as the task, only autonomous vehicles as the domain, only synthetic-to-real as the source-target setup, and only GTA as the synthesizer and city-escapes as the real-world target. Despite this clear limitation, the results are probably already interesting but the statements throughout the paper should be accordingly delimited so it doesn’t read as if it is the general video domain adaptation that is the subject of the study. Perhaps action recognition could be used and/or other datasets such as CamVid, LaRS, or VSPW to remain more general.
- there is always a conclusivity problem when it comes to “negative results”, in this case on improving image-DA with ideas transferred from video-DA. One can always posit there exists a better adaptation of those techniques. But the paper seems to have conducted the naive adaptation properly.
- I am also missing side/ablation studies shedding light into why the video domain adaptation is not effective beyond the specific image domain adaptation. For instance in what way the improvements brought about by multi-resolution possibly overshadow any possible improvement from temporal information? Do the remaining errors, after multi-resolution modeling, in the target domain have characteristics that make them difficult which are independent of the temporal information? A discussion of, at least possible hypotheses based on the results and the experiences of the benchmarking, is missing in general.

Minor points:
- The start of the abstract implicitly assumes all domain adaptations, at least for computer vision, have their images coming from video, undermining the cases where images are still and do not come as frames of videos. It is better to reword to disambiguate.
- Following are some suggested edits regarding the style of writing which although might make the paper less exciting but is probably important to follow to keep the narrative as objective as possible in an academic writing.
- - exclamation mark in the abstract and then again in 3rd paragraph of page 2 and elsewhere
- - adjective “massive” in the caption of Figure 1 and then again in 3rd paragraph of page 2 and elsewhere
- - adjectives such “startling” and “astonishing” performance in page 2 and other similar terms elsewhere.
- in sec 3.1: why should the number $N$ of source and target series in Video-DA be the same?
- in page 7, $h_\theta$ is used for both feature representation, logits, and then earlier for outputs.

---

> ### Author Response · Authors · 2023-11-08
> **Response**
>
> Thank you for the thorough review! We address specific concerns below:
>
> > there is only one line of work for Image-DA that is tested on videos.
>
> This is a valid concern! Please see our general response above for a detailed discussion.
>
> > the benchmark test-bed is quite limited: only semantic segmentation as the task, only autonomous vehicles as the domain, only synthetic-to-real as the source-target setup, and only GTA as the synthesizer and city-escapes as the real-world target.
>
> Great suggestion!  In our updated draft, we have included additional experiments on Viper->BDDVid (labeled subset of the Berkeley Deep Drive dataset for which original videos are available) and Synthia->BDDVid. Please see general response above for details.
>
> In addition, while we would also love to see this direction of work studied for tasks other than segmentation, we agree with reviewer P6Qq that this would likely be in the scope of an additional paper.
>
> > I am also missing side/ablation studies shedding light into why the video domain adaptation is not effective beyond the specific image domain adaptation.
>
> Great suggestion, thanks! We have included a discussion of this in Section 4.4 of our revised draft, and summarize in our general response above.
>
> > The statements throughout the paper (possibly including the title) should be accordingly delimited so it doesn’t read as if it is the general video domain adaptation that is the subject of the study.
>
> As recommended, we have revised our terminology in the revised draft.  We have replaced all instances of VideoDA with Video-DAS, meaning Video Domain Adaptation for Semantic Segmentation.

---

### Author Response · Authors · 2023-11-08
**General Response to All Reviewers**

We would like to thank the reviewers for their thoughtful and thorough reviews.  We especially appreciated hearing our work described as  “easy to understand and follow up on” (reviewer RrVf), and our focus on comparing image and video DA techniques as “interesting and rightly-posed.” (reviewer P6Qq). We have uploaded a revised draft with edits in red text, incorporating the reviewer's feedback.  We address concerns shared by reviewers below:

**1) All reviewers expressed that they would like to see experiments in more settings, specifically additional datasets.**  After investigating several datasets including CamVid, ACDC and Dark Zurich, we found BDD10k to be best suited for our task, due to the availability of temporally close sequential video frames. Since prior work in video DA has not studied this shift, we had to carefully design a benchmark shift considering several intricacies (detailed in the main paper). Below, we present our key results on this custom subset of BDD10k, which we call BDDVid.

VideoDAS methods on Viper/Synthia -> CS/BDDVid
| Name | Vid | ACCEL | PL | Vip$\rightarrow$CS | Syn$\rightarrow$CS | Vip$\rightarrow$BDDVid | Syn$\rightarrow$BDDVid |
|-|-|-|-|-|-|-|-|
| Source ||| None | 36.7 | 30.1| 36.6| 25.4|
| HRDA (baseline) ||| None | 65.5 | 76.1 | 49.1| 48.1|
| + Custom || x | None | **66.9** | 76.6 || 50.3 |
| + Custom ||| Consis | 65.8| 76.5| 49.7| 51.6|

As seen, across all four shifts, no single technique consistently outperforms the others. However, we do see some consistent gains across datasets by leveraging ACCEL (0.5-2.2 mIoU) and consistency based pseudo-label refinement (0.3-3.5 mIoU, see paper for full table). Note that we were only able to run  a subset of techniques for the rebuttal due to time constraints but will include the full set in the camera ready version.

**2) Reviewers P6Qq and Ep2P expressed that they would like a discussion / investigation into “why” video methods are lacking, beyond table 3.**

Why have video methods failed to exploit ImageDAS techniques (such as multi-resolution fusion)? We believe that this is because of the siloed nature of progress in ImageDAS and VideoDAS, in the absence of a common benchmark, baseline, and codebase. Our work seeks to bridge this gap.

Why are ImageDAS and VideoDAS methods not complementary? In table 4 of the main paper, we tried to understand whether video methods could be improved by incorporating critical components from HRDA/MIC such as multi-resolution fusion (MRFusion).  However, the reviewers have pointed out that we failed to explain why the improvements of MRFusion and video based approaches are not complementary. To this point we add an analysis in section 4.4: specifically, we measure the temporal predictive consistency of an HRDA model with and without MRFusion, and find that models trained with MRFusion have high temporal predictive consistency (see tables below), which in turn implies that there is very little temporal signal left for a video based approach to leverage.  One possible direction of future work might be to consider larger frame distances where the predictive consistency is not as high.

Temporal Consistency of Predictions (PL-PredConsis-IoU) with and without MRFusion.  DLV2 backbone trained on Viper to Cityscapes:
| Method      | road | side | buil | fen  | light | sign | veg  | ter  | sky  | per  | car  | truck | bus  | mot  | bic  | PL-PredConsis-mIoU |
|-|-|-|-|-|-|-|-|-|-|-|-|-|-|-|-|-|
| No MRFusion | 94.6 | 82.7 | 92.5 | 58.7 | 67.2  | 76.8 | 92.1 | 73.6 | 92.6 | 79.2 | 89.7 | 52.7  | 72.4 | 71.4 | 79.2 | 78.4                        |
| HRDA        | 98.6 | 93.3 | 96.3 | 83.3 | 80.1  | 85.2 | 95.7 | 85.6 | 97.2 | 84.4 | 96.2 | 92.1  | 94.9 | 79.4 | 85.7 | 89.9                        |

Temporal Consistency of Predictions (PL-PredConsis-IoU) with and without MRFusion.  DLV2 backbone trained on Synthia to Cityscapes:
| Method      | road | side | buil | pole | light | sign | veg  | sky  | per  | rider | car  | PL-PredConsis-mIoU |
|--|-|-|-|-|-|-|-|-|-|-|-|-|
| No MRFusion | 87.6 | 83.0 | 88.3 | 66.1 | 72.4  | 77.3 | 92.6 | 94.4 | 79.6 | 81.3  | 93.6 | 83.3                        |
| HRDA        | 96.4 | 88.1 | 94.0 | 67.6 | 75.3  | 81.2 | 95.0 | 96.7 | 81.0 | 83.7  | 95.0 | 86.7                        |

**3) Reviewers P6Qq and Ep2P expressed that our results only considered one line of work: HRDA+MIC.**

We have strengthened this part of the paper in section 3.2.  Since the advent of Daformer, practically all methods in ImageDAS have built off of this core architecture ([leaderboard](tinyurl.com/4kvxxp5p)), considerably outperforming previous adaptation paradigms (for instance TransDA-B lags behind the SOTA by 12 mIoU). Accordingly, we restrict our ImageDAS baseline to this line of work. However, we emphasize that our focus is to answer whether we are using videos correctly, and to study this we combine HRDA with a comprehensive set of video DA strategies from the literature.

---

> ### Comment · Reviewer_P6Qq · 2023-11-08
> **videoDAS results for BDD10k**
>
> Thank you for the thorough response and revision. Where can we see the videoDAS results on the subset of BDD?

---

> > ### Author Response · Authors · 2023-11-08
> > **Re: videoDAS results for BDD10k**
> >
> > It's in Table 4 (page 10) of the draft!  We included the abbreviated version in the response itself

---

> > > ### Comment · Reviewer_P6Qq · 2023-11-09
> > >
> > > What I am asking for is the VideoDAS baselines for the cases with BDD -- similar to Table 2. What I see in Table 4, as far as I understand, is the addition of VideoDAS techniques to the HRDA baseline.
> > >
> > > As an aside, not a major point but it is good to refrain from using exclamation marks in academic communications. I suggested this for the paper and it was not rectified (still exists where it was), it is used again in the added red text, and now it is used in your answer to my question. Not a good practice.

---

> > > > ### Author Response · Authors · 2023-11-10
> > > > **Re: Official Comment by Reviewer P6Qq**
> > > >
> > > > Given the time constraints we didn't think it was feasible to generate Table 2 for the Viper$\rightarrow$BDDVid and Synthia-Seq$\rightarrow$BDDVid shifts.  This would require us to modify each work's codebase to be compatible with BDDVid.
> > > >
> > > > Thank you for your feedback regarding our writing.  We have updated our draft accordingly.

---

> > > > > ### Comment · Reviewer_P6Qq · 2023-11-22
> > > > >
> > > > > Given the new results:
> > > > >
> > > > > - the new experiments do not provide new results with regards to the setup on which it is tested how the sota Video-DAS methods perform compared to HRDA. So the scope of that remains quite limited as mentioned in the original review. This is important as when one reads the title/abstract/introduction it seems this is the main message of the paper.
> > > > > - regarding the secondary message of the paper: the new experiments does add to the diversity of the setups where it is tested how much the established Video-DAS techniques can add on top of the sota Image-DAS, i.e. HRDA. The new results, in fact, show that ACCEL and PL can be helpful although not by a large margin.
> > > > >
> > > > > Considering the above two points beside the fact that HRDA does also outperform prior Image-DA methods by a large margin, I think the statements in the paper can still be finessed not to convey a stronger message than the experiments can corroborate.
> > > > >
> > > > > As my original review suggests, the paper has a point and relevant audience and the released codebase will be useful for future works, both on Image-DAS and Video-DAS but there needs to be an adjustment between the claims and the experiments. I understand that doing more experiments can be prohibitively demanding and that is the authors' choice as there is enough contribution as is, so I suggest the writing is further adjusted as necessary to align with what experiments are available.

---

> > > > > > ### Author Response · Authors · 2023-11-22
> > > > > > **Re: Official Comment by Reviewer P6Qq**
> > > > > >
> > > > > > > Considering the above two points beside the fact that HRDA does also outperform prior Image-DA methods by a large margin, I think the statements in the paper can still be finessed not to convey a stronger message than the experiments can corroborate.
> > > > > >
> > > > > > Thank you for the feedback, we will appropriately moderate our claims to align with the experiments we have.

---

> > > > > > > ### Author Response · Authors · 2023-12-07
> > > > > > > **Re: Official Comment by Reviewer P6Qq**
> > > > > > >
> > > > > > > We have submitted our final revision, with edits in green.  Please let us know if there's any additional feedback, and we'd be happy to address it.  Thank you for your help.

---

### Decision · Action_Editor_kGLD · 2023-12-26

**Recommendation:** Accept as is

**Comment:**

The AE sides with acceptance. The work is informative and the review and revision process has brought the claims and evidence into better agreement. The chosen methods and benchmarks are state-of-the-art (or near state-of-the-art) and popular, so these results can guide further examination of image and video DA, because the experimental study is thorough within its chosen breadth. Nevertheless, the AE must echo the feedback from two of three reviewers that a general claim along the lines of "not using video effectively" would ideally be justified by evidence from more datasets and tasks. The degree of generality or lack thereof was a key point in the official recommendations, and the cause for diverging scores in leaning toward acceptance vs. rejection. While the AE respects the disagreement raised, this is not an obstacle to acceptance, because the intended scope of the paper is well-signaled and gaps have been adequately addressed by the revision. In summary, the positives for informing the community with the experimental comparison and analysis overcome the negatives w.r.t. a potentially even broader paper. The paper can be accepted as is.

Regarding certifications, 2/3 reviewers have voted for the reproducibility certification. Given the level of detail in the submission and the linked code, the AE agrees it is reproducible. To quote the [editorial policies](https://jmlr.org/tmlr/editorial-policies.html#certifications):

> Reproducibility Certification. This is awarded to papers whose primary purpose is reproduction of other published work. Beyond simple verification, the paper must contribute significant added value through additional baselines, analysis, ablations, or insights.

While I would say the _primary_ purpose of this paper is analysis and benchmarking, rather than reproduction, this work does reproduce a number of methods and benchmark settings and certainly provides added value. Therefore I will leave this to the Editors to confirm.

The AE thanks the authors and reviewers for engaging in the TMLR process to deliver an improved paper with agreement between its claims and evidence and a clear audience.

As a last note, happy winter holidays to you all!

**Audience:**

All reviewers agree there is an audience. The audience for this work includes those interested in domain adaptation, semantic segmentation with reduced annotations, and video processing. These topics are highlighted in the title and abstract so the work appropriately identifies itself to this audience. Since these subjects are current topics of interest at the intersection of vision and learning, this work is certainly in scope for TMLR. The specific cross-pollination goal of this work bridges image-based and video-based adaptation and is informative to both sides. While such an audience is sufficient, the AE does note that a wider audience would be reached by considering more methods (especially for image DA) and more tasks (such as detection or action recognition) given the generality of the title in "using videos effectively".

**Claims And Evidence:**

Two of three reviewers lean toward acceptance while one reviewer leans toward rejection. All reviewers agree that there is interest in providing a comparative study of image vs. video DA and that the experiments sufficiently cover datasets, video DA methods, and ablations to be empirically informative. However, there is disagreement on if the scope of experiments is sufficient, in particular the number of image DA methods and datasets, and if all of the quantitative results are significant enough to justify the claims of differences (although certainly some of the margins are large indeed). While a more comprehensive experimental study would provide more evidence still, and while more analysis might shed more light on how image and video DA differ to explain their accuracy discrepancies, this work delivers an informative experimental study within the scope it has claimed for itself. Furthermore the revision has improved the submission by expanding its scope to more benchmarks and methods, analyzing the combination of image and video DA methods, and discussing potential reasons behind the ineffectiveness of video DA for improving on image DA. The writing has also been improved to more exactly qualify the scope of the paper, to make sure that is reflected in its terminology, and to moderate claims about the generality of the experiments concerning image vs. video DA.

---

> ### Author Response · Authors · 2024-01-28
> **Thank you**
>
> We have submitted our camera ready version.  Thanks for a smooth review process, and hope you had a great winter holiday as well